# Alternative Cell Sources for Liver Parenchyma Repopulation: Where Do We Stand?

**DOI:** 10.3390/cells9030566

**Published:** 2020-02-28

**Authors:** Tine Tricot, Jolan De Boeck, Catherine Verfaillie

**Affiliations:** Stem Cell Institute Leuven, KU Leuven, 3000 Leuven, Belgium; tine.tricot@kuleuven.be (T.T.); jolan.deboeck@kuleuven.be (J.D.B.)

**Keywords:** hepatocyte transplantation, preclinical mouse models for liver damage, primary human hepatocytes, hepatocyte expansion, iPSCs, MSCs

## Abstract

Acute and chronic liver failure is a highly prevalent medical condition with high morbidity and mortality. Currently, the therapy is orthotopic liver transplantation. However, in some instances, chiefly in the setting of metabolic diseases, transplantation of individual cells, specifically functional hepatocytes, can be an acceptable alternative. The gold standard for this therapy is the use of primary human hepatocytes, isolated from livers that are not suitable for whole organ transplantations. Unfortunately, primary human hepatocytes are scarcely available, which has led to the evaluation of alternative sources of functional hepatocytes. In this review, we will compare the ability of most of these candidate alternative cell sources to engraft and repopulate the liver of preclinical animal models with the repopulation ability found with primary human hepatocytes. We will discuss the current shortcomings of the different cell types, and some of the next steps that we believe need to be taken to create alternative hepatocyte progeny capable of regenerating the failing liver.

## 1. Introduction

The remarkable regenerative capacity of the liver has been appealing to human imagination since centuries. In Greek mythology, the Greek gods exploited the regenerative capacity of the liver when they punished Prometheus by having an eagle eat his liver, which grew back overnight. Nevertheless, the cellular and molecular mechanisms that underlay liver regeneration remain poorly understood.

The liver plays a key role in many life-supporting metabolic functions (e.g., bile production, glucose metabolism and glycogen storage, vitamin storage, metabolization of drugs and other xenobiotics, production of albumin and coagulation factors, etc.). Therefore, failure of restoring basic hepatic functions can have devastating consequences and can become life threatening. Interestingly, and relatively unique to the liver, the first line of defense against liver damage is that mature hepatocytes, that are chiefly in G_0_, re-enter the cell cycle to replace the lost hepatocytes, or undergo hypertrophy. However, if hepatocytes are damaged too extensively to participate in the liver repair process, liver regeneration occurs via differentiation of (facultative) hepatocyte progenitor/stem cells. The origin of these progenitors remains not fully understood, and likely includes liver progenitor cells, cholangiocytes, and intermediary hepatocytes (recently reviewed in [1]). However, if liver damage persists, or gives rise to liver fibrosis and cirrhosis, none of the above mechanisms can repair lost hepatocytes, giving rise to liver failure. In such cases, orthotopic liver transplantation (OLT) is currently accepted as the gold-standard treatment. OLT is also the treatment of choice for certain inborn liver-based metabolic diseases (reviewed in [2]). Even though OLT can provide a cure, long waiting lists related to shortage of suitable donors, the morbidity associated with this therapy, the high cost of the treatment, and the need for life-long immunosuppressive treatment are some of the reasons why alternative therapies are sought after.

One such alternative for OLT is transplantation of individual cells. Cells already used clinically are primary human hepatocytes, isolated from cadaveric livers, to regenerate the liver tissue. Other cells, in clinical trials or tested in preclinical studies, include hepatocyte progenitors or expanded liver derived cells, but also cells from tissues other than liver, like mesenchymal stromal cells, and transdifferentiated or dedifferentiated cells that are then redifferentiated to cells with hepatocyte features. Mechanisms of action of these latter cells include direct regeneration, but also fusion with existing hepatocytes and—as a result of this—in vivo transdifferentiation into hepatocytes [3,4], or support of regeneration via secretion of paracrine factors [5,6]. In this review, we will focus chiefly on which (stem) cell populations are candidate cells for engraftment and repopulation of the liver, based on preclinical and clinical studies. The definitions we will be using to describe the different effects exerted by transplanted cell populations on the liver are detailed in Table 1.

## 2. Animal Models Used to Investigate Hepatocyte Transplantation in Liver

### 2.1. Animal Models

In the past decades, the repopulation potential of human hepatocytes has been extensively studied by making use of small animal models, mainly rodents, but also larger animal models [9]. To allow engraftment of hepatocytes in the host liver, severely immunodeficient recipient animals are needed, often combined with chemical, surgical, or genetic interventions, to allow selective growth advantage of the human hepatocytes over the host hepatocytes. Several experimental models have been used to induce such damage:

#### 2.1.1. Partial Hepatectomy

Initially, hepatocyte transplantation was studied in rodents that underwent a partial hepatectomy. Partial hepatectomy is a well-tolerated and well-controlled procedure in which typically two thirds of the liver tissue is excised, which permits transplanted hepatocytes to proliferate and repopulate the damaged rodent liver [10].

#### 2.1.2. Chemical Toxin Induced Hepatocyte Loss

Next to partial hepatectomy, chemical toxins such as carbon tetrachloride (CCl_4_), retrorsine, dimethylnitrosamine, monocrotaline, or acetaminophen (APAP) have been used to cause severe hepatotoxicity in rodent models. Unlike partial hepatectomy, use of chemical injury provides for a less controlled and targeted liver repopulation model, as the chemical can also cause inflammation due to infiltration of immune cells into the necrotic area to remove the apoptotic hepatocytes [10,11].

#### 2.1.3. Transgene-Induced Liver Damage Mouse Models

An alternative is to use transgene-induced liver damage mouse models to permit more efficient human hepatocyte repopulation. The first established model was the urokinase plasminogen activator–severe combined immunodeficiency (*uPA*^+/+^-SCID) mouse model [12]. In these mice, the *uPA* gene, under control of the albumin promoter, is constitutively expressed in hepatocytes, causing hepatic injury. This allows selective expansion of human hepatocytes upon transplantation. Human hepatocytes are typically injected at ±1 month after birth. Given the young age of the animals, the transplantation procedure is stressful, and animals only tolerate cell doses of about 5 × 10^5^–1 × 10^6^ cells. Moreover, only homozygous uPA-SCID recipients allow for efficient and stable engraftment of exogenous hepatocytes, hence strongly reducing the number of experimentally available pups within one litter. Tateno at al. developed the so-called cDNA-uPA-SCID model, a mouse model expressing the cDNA of *uPA* (instead of the whole gene), which allows repopulation of human hepatocytes in hemizygous animals [13]. In addition, uPA overexpressing mice can be backcrossed with immunodeficient mouse strains other than SCID mice, like *Rag2^−^/^−^* [14,15] and *Rag2^−/−^IL2rg^−/−^* [16].

A second mouse model, the fumaryl acetoacetate hydrolase (*Fah)^−/−^Rag2^−/−^IL2rg^−/−^* (FRG) mouse, is based on the genetic knockout of the *Fah* gene. FAH is essential in the tyrosine catabolic pathway and absence of the protein results in the accumulation of fumaryl acetoacetate in the murine hepatocytes, leading to hepatic damage. This can be prevented by treating the mice with 2-(2-nitro- 4-trifluoromethylbenzoyl)-1,3-cyclohexanedione (NTBC), which blocks the accumulation of fumaryl acetoacetate by inhibiting 4-hydroxyphenylpyruvate-dioxygenase. Withdrawal of NTBC typically results in demise of the animal within 4–8 weeks. Even after transplantation of healthy hepatocytes, animals should be provided with NTBC intermittently, until the donor cells have sufficiently repopulated the liver to support hepatic functions [17,18]. As controlled administration of NTBC can keep the animals alive and injury can be induced at any moment by withdrawal of the drug, older animals and, therefore, higher cell doses (up till 5 × 10^6^) can be used. Similar to *uPA* overexpressing mice, FRG mice can be backcrossed with animals of different immunological backgrounds, e.g., the non-obese diabetic (NOD) strain (FRGN mice) and the severe-combined immunodeficiency (SCID) strain (FRGS mice) [19,20,21].

Additional transgenic mouse models have been developed, of which the thymidine kinase TK-NOG [22,23] and the AFC8 [24,25] mouse models are the most prevalent. TK-NOG mice are immunodeficient mice that express herpes simplex type 1 thymidine kinase in hepatocytes following ganciclovir administration, which causes hepatotoxicity [22,23]. However, repopulation in TK-NOG mice is less efficient than in uPA-SCID mice [22]. The AFC8 mouse contains an FK506 binding protein-Caspase 8 fusion gene under control of the albumin promoter. When dimerizer ligand AP20187 is administered, the fusion protein dimerizes, leading to apoptosis of the hepatocytes [24,25]. The AFC8 model, developed by Washburn et al., in BALB/c *Rag2^−/−^-γC^−/−^* mice, was the first humanized dual chimeric mouse model with both a humanized liver and immune system. However as for TK-NOG mice, repopulation of human hepatocytes in AFC8 mice has been reported to be less efficient than in uPA-SCID and FRG mice [24].

#### 2.1.4. Additional Methods to Improve Engraftment Efficiency

Additional methods to improve liver repopulation potential in rodents have been investigated. This includes the use of additional immunosuppression, such as administration of the Natural Killer Cell inhibitor, anti-asialo-GM1, or an inhibitor of human complement activation, Futhan [13,26,27]. Mice have also been treated with JO2, a mouse CD95 antibody that increases apoptosis of mouse liver cells and enhances human hepatocyte engraftment. Alternatively, it might also be possible to enhance the engraftment efficiency of the grafted hepatocytes by treatment with, e.g., XMU-MP-1, an inhibitor of pro-apoptotic MST1/2, or the 5D5 antibody, an agonist of the c-Met receptor and therefore an inducer of hepatocyte proliferation. Both molecules are known to promote cell repopulation and have been used by Yuan et al. to further increase the liver repopulation efficiency of engrafted hepatocytes in FRGS mice [20,21].

### 2.2. Read-Out of Engraftment and Repopulation

The engraftment and repopulation of human hepatocytes in rodent livers is quantified by different experimental tools. The presence of human cells can be detected via PCR for human specific DNA/RNA sequences. To more accurately quantify the engraftment efficiency and repopulation index (RI), immunohistochemical analyses of liver sections or flow cytometry are required. Typically, immunostaining for markers such as human albumin (hALB), human mitochondria, human sodium-taurocholate co-transporting polypeptide (hNTCP), human α1-antitrypsin (hAAT), etc., are performed. However, in FRG mice, the RI can easily be monitored by staining for FAH, as FAH^+^ cells can only originate from grafted cells in this model. Furthermore, ELISA for hALB in mouse serum is a measure for engraftment of human hepatocytes. As there is a correlation between levels of hALB in the host serum and RI of the human hepatocytes, this test also approximates the level of repopulation in a non-invasive manner. Some of these readouts can be confounded by problems such as cross-reactivity of antibodies and/or primers with mouse hepatocytes. Therefore, studies should employ proper controls to eliminate the possibility of also inadvertently measuring mouse hepatocytes, and multiple independent assays should be used to ensure the accurateness of the read-out and its interpretation. Additionally, one must also keep in mind that cell fusion might be happening rather than direct repopulation, especially in FRG mice, and this must also be tested. Finally, infecting the liver with human hepatotropic viruses such as hepatitis B virus (HBV) and hepatitis C virus (HCV) can also be used to confirm the human origin of the hepatocytes.

## 3. Grafting of Gold-Standard Control, Freshly Isolated Primary Human Hepatocytes

In 1969, Berry and Friend described a collagenase-based perfusion protocol which enabled them to efficiently isolate primary hepatocytes from rat livers [28]. As this discovery made it possible to isolate viable single cells from hepatic tissue, it laid the foundations for the subsequent development of hepatocyte transplantations. Almost one decade later, Groth et al. were the first to perform an actual transplantation of isolated primary rat hepatocytes into the liver of Gunn rats [29].

Over the years, several adaptations of the collagenase perfusion method led to the application of this protocol for the isolation of primary human hepatocytes (PHHs) [30], whose repopulation potential has been extensively studied in rodents. Transgene-induced liver damage mouse models have been used to evaluate the repopulation of PHHs in mouse liver. In uPA-SCID mice, hepatocyte loss is relatively mild, which is seen from the relative high survival rates of non-chimeric animals, but is protracted, allowing efficient repopulation by human hepatocytes [26,31,32]. Meuleman et al. obtained stable engraftment of human hepatocytes that were successfully infected with HCV for at least 4 months with RI ranging between 25% and >90%, depending on the source of PHHs [31,33,34]. Nevertheless, humanized uPA-SCID mice are not as healthy as their littermates transplanted with murine hepatocytes, suggesting that human grafts do not necessarily affect the overall long-term health of the recipient animals in a positive manner [31]. Similarly, Tateno et al. reported RIs up to 96% in uPA-SCID mice treated with anti-asialo GM1 antibodies and Futhan 6 months after transplantation [26]. Omission of anti-asialo GM1 antibodies and Futhan caused significant mouse mortality within 2 months even if the RI was >50%. Tateno et al. also successfully transplanted PHHs in hemizygous cDNA-uPA-SCID mice, achieving RIs similar to those observed in uPA-SCID mice for at least 6 months.

PHHs can also successfully repopulate the liver of FRG mice. In the original report, 1 × 10^6^ PHHs were transplanted in FRG mice wherein the *uPA* gene was overexpressed using adenoviral vectors [17]. This yielded RIs in ±1/3rd of recipients of >1% and RIs of >30% in 16% of recipients where human albumin levels of >1 mg/mL were found, and this after only 1 round of NTBC re-administration. Subsequent studies demonstrated similarly high levels of repopulation even without uPA adenoviral vector infusion, when 5 × 10^6^ PHHs were grafted [35]. Moreover, human hepatocytes, harvested from primary transplanted FRG mice can be highly successfully re-transplanted in secondary FRG mice [36], effectively expanding human hepatocytes in mice in vivo. This has enabled the routine use of mice with a humanized liver (Yecuris FRG and FRGN mice) for studies related to viral infections, gene therapy, drug metabolization, and disease modeling [36].

In addition, human hepatocyte repopulation in other transgene-induced liver damage mouse models, such as TK-NOG and AFC8 mice, was also demonstrated, even if RIs appear lower in these mice than uPA-SCID or FRG mice. Hasegawa et al. observed repopulation of ±43% in TK-NOG mice [22], while Washburn et al. demonstrated a RI of ±15% in AFC8 mice [24] (of note, cells grafted in AFC8 mice were not PHHs but hepatocyte progenitor cells).

Both freshly isolated as well as cryopreserved PHHs have been used to graft in mice [26,31]. However, cryopreserved PHHs often have a poorer cell viability compared to freshly isolated PHHs, and their functionality and plating ability are significantly poorer compared to freshly isolated PHHs [37,38,39]. Nevertheless, successful transplantations have been performed with cryopreserved PHHs and it should be noted that the use of cryopreserved PHHs is more user-friendly in a preclinical setting [26,39].

The observation that human hepatocytes could repopulate mouse livers led to the first transplantation of primary hepatocytes in 10 patients in 1992 [40]. Since then, several reports have been published describing PHH transplantation in patients with a number of different liver diseases (reviewed in [41]). Despite these promising (pre)clinical results obtained with PHHs, the bottleneck in the whole process persists, namely, the shortage of suitable donor material. In preclinical mice models, serial transplantation can be used to overcome this problem. As described above, isolation of donor hepatocytes from a primary host and subsequent transplantation of these hepatocytes into a secondary host enables the expansion of fresh PHHs from the same source and the generation of a large number of transplanted animals from a single donor [17]. However, this is not clinically applicable. Therefore, researchers have been investigating strategies for expanding the limited pool of PHHs or generating de novo hepatocytes from other cell sources suitable for downstream applications (Figure 1).

Finally, it is of importance to note that mature (adult liver-derived) hepatocytes are the true gold-standard cells for liver repopulation. It is, indeed, generally accepted that fetal hepatocytes have a far less engraftment/repopulation potential than their mature counterparts [42,43,44,45], likely because they are less capable to home and subsequently expand in the adult cell niche of the host liver. Therefore, in the search for alternatives for PHHs, it will likely be necessary to use/create cells with functional characteristics that approximate those of fully mature hepatocytes.

## 4. Strategies to Create Larger Numbers of Hepatocytes from Primary Liver Tissue

### 4.1. 2D Cultured Hepatocytes

Since the beginning of this century, multiple protocols have been developed wherein PHHs are expanded using growth factors and small molecules, but without immortalization, and engraftment of such expanded hepatocytes in rodent livers has been tested. It should be noted that these expansion protocols lead to partial dedifferentiation of the PHHs to cells with more progenitor-like features, which can be redifferentiated often to biliary and hepatic progeny. A few recent examples with quite variable repopulation outcomes are described here. Unzu et al. developed a chemically defined protocol to efficiently expand PHHs. When passage 3 expanded hepatocytes were grafted in the spleen of FRG-NOD mice, minimal engraftment in the liver was seen, and cells were retained in the spleen, and this in stark contrast to the uncultured PHHs [46]. Kim et al. could also expand PHHs more than 150-fold using defined factors. When grafted in three different mouse models, ±20% engraftment of expanded hepatocytes (passages 4–6) was observed in mouse livers, and human cells were detected for at least 8 weeks [47]. Comparisons with the original unexpanded PHHs were not done. Finally, expanded PHHs described by Fu et al. appeared to have similar RIs (maximum RI of 16% after passage 5) as non-expanded PHHs (maximum RI 25%) when grafted in anti-asialo-GM1 treated *Fah^−/−^Rag2^−/−^* mice, also treated with the pharmacological immunosuppressant FK506 [48]. However, this required transplantation of 2 × 10^6^ cells/mouse, which is much higher than in the original paper describing grafting PHHs in this mouse model [44].

Thus, although repopulation of in vitro expanded human hepatocytes between 80 and 90% has been reported in some individual mice in the studies above and other studies [49,50], in general a drastic reduction in repopulation potential is observed following several passages (typically around 6 passages in the cited references) of PHHs (as hepatic progenitors) in 2D culture systems. Hence, expansion of PHHs is only partially capable of overcoming the limiting availability of human hepatocytes.

### 4.2. Liver Progenitor Cells 

Consistent with the studies above where, following in vitro expansion, PHHs dedifferentiate to progenitor cells that can still repopulate mouse livers to some extent, grafting of (both fetal and adult) liver progenitor cells (LPCs) also results in liver repopulation. Although less mature cells are believed to engraft and repopulate a host liver less efficiently than mature hepatocytes [16,42,44,45], progenitors have been proven to be useful for transplantations. This has been demonstrated not only in animal models (both single [16,42,45,51] and serial transplantations [44] in transgene-induced liver damage mouse models; in general also treated with Futhan, anti-asialo GM1 or oncostatin M (OSM) to increase efficiency), but also clinically. Sokal et al. transplanted 9 × 10^8^ adult mesenchymal-like liver progenitor cells (0.75% of total body mass) in a 3-year-old female patient with ornithine carbamoyltransferase (OTC) deficiency, via a percutaneous intraportal catheter. Fourteen weeks after transplantation, liver biopsies showed a RI of 3–5% of donor cells. However, no long-term follow up analysis was performed as the child underwent OLT 6 months after the LPC transplantation [52]. These results are in line with murine studies wherein LPCs are grafted. Longer-term follow-up will be needed to assess if LPCs can reach RIs of 5 to 10%, generally considered as being necessary to have a long-term therapeutic effect in patients [7]. Nevertheless, phase I/II clinical trials have been performed using LPCs to treat patients with an urea cycle disorder and Crigler–Najjar syndrome [53]. Unfortunately, as LPCs are even less abundant than PHHs, they will not be able to solve the shortage of donor material for transplantation, unless they can be extensively expanded ex vivo without loss of repopulation ability.

### 4.3. Liver Organoids

Liver organoid cultures are a promising new addition to the arsenal of cultured hepatocytes. Organoids are 3D structures that originate from adult, organ specific stem cells, and develop into a collection of organ-specific cell types by differentiation and self-organization [54]. Although many different subpopulations of progenitor cells have been identified in the liver, the progenitor cells used for liver organoids are typically characterized as EpCAM^+^ ductal cells [55]. In 2015, the Clevers group first reported on the creation of long-term expandable human liver organoids from EpCAM^+^ cells. They demonstrated that organoid-derived cells (i.e., from organoids of passage 6–10) can engraft into the liver of immunodeficient mice after treatment with a mix of retrorsine, anti-asialo GM1 and CCl_4_, followed by in vivo differentiation towards hepatocytes [27]. Histological examination of the liver of these mice, identified small clusters of human hepatocytes, even if the degree of repopulation was not quantified. Consistently, low levels of hALB (50–100 ng/mL) could be detected in murine blood for 3 months after grafting. This was similar to levels found following grafting 1–2 × 10^6^ PHHs, even if these levels are much lower than observed in other studies, albeit in different mouse models [13,31,32]. This initial study suggested thus that liver organoids engraft, but do not truly repopulate the liver of this mouse model. Since then, this group has optimized creation of organoid cultures from human liver. In a recent study, they generated Hep-Orgs, that did not re-express EpCAM upon culturing. Hep-Orgs were generated from adult, pediatric (~7 months), and fetal (11–20 weeks) livers [42]. The latter could be expanded for over 11 months (i.e., 28 passages) and could, even after 16 passages, be differentiated in vitro by dissociating and replating in 2D collagen-coated plates. After an in vitro maturation step, 1–3 × 10^5^ progeny of these Hep-Orgs (from a passage 16 fetal liver, or from a ±6 passage pediatric liver (described in the manuscript as following 2-month expansion, and passaged every 7–10 days)) were grafted in the spleen of FRGN mice, also preconditioned with retrorsine, OSM, and adenoviral overexpression of human hepatocyte growth factor (HGF). This resulted in hepatocyte engraftment in murine livers, with appearance of large clusters of FAH^+^ cells over a period of several months, indicating repopulation. However, the precise repopulation degree was not calculated. Consistently, a progressive increase in hALB in mouse blood was found, to levels which were, however, lower than in mice transplanted with matched fetal or pediatric PHHs. Although other groups have also reported on liver organoids (e.g., in [56]), we are not aware of studies wherein progeny of these organoids were grafted in vivo.

Thus, the expansion of liver progenitor cells using liver organoid cultures can also create progeny that is capable of engrafting and (to some extent) repopulating mouse livers. However, it is not clear whether the repopulation seen, starting from organoids, outperforms progeny from 2D cultured hepatocytes, and direct head-to-head comparisons might be enlightening. As repopulation by progeny from 3D organoid cultures, which are relatively labor intensive to generate, even if expanded and repopulated to >16 passages (for fetal liver derived organoids), is possible, they currently appear to not yet be a suitable source to replace PHHs for liver repopulation.

## 5. Generating Hepatocytes from Tissues Other Than Liver

Cells from tissues other than liver have also been tested for their ability to engraft and repopulate the liver or support failing livers in some other way.

### 5.1. Hematopoietic Stem Cells

Since 2000, a number of studies demonstrated that both animal and human bone marrow-derived cells might hold the potential to generate hepatocytes in vivo [57,58]. Follow-up studies attributed this capacity to hematopoietic stem cells (HSCs) [59] and later studies pin-pointed lineage-committed granulocyte-macrophage progenitors (MGPs) and bone marrow-derived macrophages (BMMs) as important sources of such bone marrow-derived hepatocytes (BMHs) [60]. Different reports provided evidence that the principal mechanism of generating BMHs consists of fusion between host hepatocytes and the transplanted bone marrow cells, in both animal models (RIs of more than 30%) [3,61] and human patients (RI 1–2%) [62,63], although direct conversion of HSCs into hepatocytes has also been proposed (early engraftment with RI of 6–7%) [64].

Even though high RIs can be achieved in FRG mice using bone marrow transplantations, the successful outcome strongly depends on the very high selective pressure of multiple rounds of NTBC cycling, which enables the small number of fused cells to expand sufficiently to repopulate the liver over protracted time periods [63]. As such highly selective pressure cannot be achieved in human patients, clinical applications for treating liver diseases based on HSC transplants are unlikely, and have, to the best of our knowledge, not been reported yet.

### 5.2. Mesenchymal Stromal Cells

A clinically relevant cell type, at least when considering liver regeneration, is mesenchymal stromal cells (MSCs). MSCs are multipotent adult stem cells that reside in many different tissues, like bone marrow, fat tissue, lung, liver, and the umbilical cord. MSCs have been used (1) without prior manipulation towards hepatocytes before grafting or (2) following differentiation in vitro towards the hepatic lineage.

#### 5.2.1. Undifferentiated Mesenchymal Stromal Cells

Several reports have shown that transplantation of human MSCs of different origins (e.g., umbilical cord [6,65,66], amniotic mesoderm [67,68], bone marrow [9,69,70,71], liver [72], etc.), without prior hepatic commitment can support liver function and/or alleviate symptoms of liver failure in both animal models as well as human patients [73,74,75]. The molecular mechanisms underlying these beneficial effects are manifold, but are believed to be derived from paracrine factors affecting apoptosis, immune responses, stellate cell activation, angiogenesis, etc. (as reviewed in [76,77,78]). Moreover, as for BM-HSCs and BMMs, fusion of human MSCs with host liver cells has been suggested [79].

Although, strictly speaking, no actual engraftment of MSCs in the host liver is required for exerting their strong immunomodulatory effects and relief of acute liver failure [6,70,80], these cells are able to do so, with (according to some reports [9,65,69,72]) subsequent spontaneous in vivo differentiation into cells with some hepatic features. Depending on several experimental parameters, including the experimental model and downstream read-out, highly variable outcomes of MSC transplantation in mouse liver can be found in the literature. For example, Yuan et al. reported stable engraftment (RIs between 45% and 50%, 60 weeks after intrasplenic injection) of human bone marrow-MSCs (BM-MSCs), in *Fah^−/−^Rag2^−^**^/−^IL2rg^−/−^* SCID (FRGS) mice treated with JO2 antibody and in vivo differentiation towards hepatic cells [69]. Such high RIs are in strong contrast with other reports on direct transplantation of MSCs. Shi et al. only observed ±4.5% engraftment of the same cell type in a pig model of D-galactosamine induced fulminant hepatic failure [9], and Baertschiger et al. could not find back any BM-MSC-derived hepatocytes in their NOD-SCID mice after intrasplenic injection [81]. The use of FRGS mice by Yuan et al. together with the very long-time periods before analysis (i.e., 12–60 weeks) would be consistent with occurrence of fusion events. However, the authors assessed for fusion by double staining for murine and human major histocompatibility complex (MHC) and were unable to find double positive cells.

Thus, undifferentiated MSCs are currently being tested in clinical trials to support hepatic function and/or reduce liver fibrosis. However, the consensus is that the mechanism(s) underlying these effects are the strong immunomodulatory and anti-inflammatory effects associated with MSCs, and to a lesser extent trophic effects on hepatocytes, not the repopulation of the liver parenchyma [73,74,75]. In addition, these cells typically originate from organs which are difficult to access, like bone-marrow or fat tissue, and sources like the umbilical cord and amniotic membrane that cannot be obtained from autologous sources, possibly necessitating immunosuppressive treatment of the patient in the setting of transplantation, even if it is possible that due to the immunomodulatory characteristics of MSCs grafting across human leukocyte antigen (HLA) barriers may be possible without immunosuppression [82,83]. Of note, however, even if we believe that undifferentiated MSC are not good candidates for liver repopulation, as there are only few studies describing spontaneous differentiation and repopulation after engraftment, they might hold potential for grafting together with hepatocytes from other cell sources that have the potential to regenerate the liver, because of their ability to protect the latter cells from immune rejection. In addition, undifferentiated MSCs may ameliorate liver fibrosis [76], even if there are also reports suggesting that they may aggravate fibrogenesis [81,84], making the case for in vitro differentiation prior to transplantation.

#### 5.2.2. Hepatic Differentiated Mesenchymal Stromal Cells

Aurich et al. demonstrated enhanced engraftment of hepatocyte-like cells (HLCs) derived from MSCs via in vitro differentiation compared with their undifferentiated counterparts. They grafted both undifferentiated MSC and MSC-HLCs via intrasplenic injection into immunodeficient *Pfp/Rag2^−/−^* mice, also treated with monocrotaline treatment and partial hepatectomy. They could observe some cells positive for hALB, and low levels (±150 ng/mL) of hALB blood only in mice grafted with the differentiated MSCs [85].

Since then, a large number of studies tested the possibility of repopulating the liver with in vitro hepatic differentiated MSCs. To generate hepatocytes from MSC, a multitude of approaches has been used including the addition of cytokine cocktails [65,86,87,88,89,90,91,92,93,94,95], small molecules [96], and/or microRNAs (miRNAs) [97,98,99] to the cell culture medium and recreation of the physicochemical characteristics of a hepatocyte niche micro-environment [100,101]. Generally, HLCs derived from MSCs possess some mature and fetal hepatocyte features as has been shown by transcriptome and functional studies [102,103]. It should be noted that some differences are also reported depending on the origin of the MSC population (e.g., BM vs. umbilical cord vs. liver-derived MSCs; pediatric vs. adult MSCs) [87,95,104].

In contrast to fresh or cultured PHHs, LPCs, or liver-organoids, MSC-HLCs at doses between 5 × 10^5^ and 3 × 10^6^ cells are commonly grafted by injection in the tail vein or via direct transplantation into the liver. In general, wild-type or immunodeficient mice without liver damage inducing transgene were used, and liver injury was caused by treatment with CCl_4_. A number of reports demonstrated presence of single cells or small groups of hALB positive cells throughout the liver (RI between 1 to 5%) 1 to 14 days after transplantation [86,89,97,98,100,105]. Most studies did not perform long-term follow-up analysis. Therefore, it remains to be determined if MSC-HLCs can repopulate mouse liver, even if Campard et al., El-Kehdy et al., and Aurich et al. demonstrated low engraftment potential of human cells between 6 to 10 weeks post transplantation [65,85,105].

As the animal models and the route of administration of MSC-HLCs differed considerably with the studies wherein PHHs or LPCs were grafted, a direct comparison is not possible. Nevertheless, currently, there is relatively little evidence that MSCs committed to the hepatic lineage in vitro form a valid alternative for PHHs to repopulate the host liver.

### 5.3. Amniotic Epithelial Cells

Aside from MSCs, the amniotic membrane also contains amniotic epithelial cells (AECs). As the amnion is derived from the epiblast during fetal development, it is believed that AECs may harbor the potential to differentiate into the three germ layers, but this without tumorigenicity and immunogenicity [106,107]. There is evidence that AECs, like MSCs, improve liver function at least in part via paracrine effects [108]. However, other studies demonstrated that AECs (both undifferentiated and differentiated towards HLCs) may be able to engraft into the liver parenchyma.

Manuelpillai et al. demonstrated that grafting 2 × 10^6^ cells cultured human AECs via tail vein into the liver of CCl_4_ treated immunocompetent mice resulted in engraftment 2 to 4 weeks after injection (grafted cells were hALB^+^, but only some expressed *hepatocyte nuclear factor 4α* (*HNF4α*)), and a reduction in liver fibrosis was observed, even if AECs induced a humoral immune response [109]. Studies also demonstrated that grafting AECs can support liver function and mitigate certain inborn metabolic diseases, like Hurler syndrome [110], Maple Syrup Urine Disease [111], and Niemann-Pick type 1 [112]. However, typically engraftment efficiencies of 0.1–5% were reported up to 6 months after injection [111,113]. In addition, pre-differentiation of AECs to HLCs in vitro has been reported. For instance, Liu et al. used a 14-day cytokine-based differentiation protocol to obtain HLCs from human AECs in 2D culture. Following grafting 2 × 10^6^ of such AEC-HLCs in CCl_4_ treated NOD-SCID mice, human cells were detected in the recipient liver (although the RI was not quantified) and liver injury was attenuated [104]. However, it remains to be addressed whether naïve AECs or in vitro differentiated AEC-HLCs are superior as an alternative cell source for alleviating liver failure and/or repopulating the host liver [114].

### 5.4. Transdifferentiation of Fibroblasts Towards Hepatocytes

Already in the 1990s, it was demonstrated that the muscle transcription factor (TF), MYOD, could transdifferentiate non-myoblasts into myoblasts [115]. This, together with the development of induced pluripotent stem cells (iPSCs), based on overexpression of TFs by the Yamanaka team [116], opened a new field of research towards creating of TF guided cells with hepatocyte function from non-hepatic cells that might rescue a diseased liver. This could be done using autologous cells that can easily be harvested, such as fibroblasts.

In these transdifferentiation studies, the hepatocyte phenotype is induced by overexpressing different sets of TFs, combined with hepatocyte supportive media. Two possible avenues are used: (1) somatic cells can be directly transdifferentiated towards hepatocytes [117,118,119], or (2) somatic cells are first partially dedifferentiated, followed by differentiation into the hepatic lineage [120,121,122].

#### 5.4.1. Direct Transdifferentiation of Fibroblasts Towards Hepatocyte-Like Cells

Huang et al. were the first to report that fibroblasts could be differentiated into HLCs by lentiviral overexpression of *hepatocyte nuclear factor* 1*α (HNF1α)*, *HNF4α,* and *forkhead box protein A3* (*FOXA3)*. These fibroblast-HLCs resembled PHHs, even though drug metabolization abilities were significantly lower than that of PHHs. Overexpression of SV40 Large T Antigen (LT) into the fibroblast-HLC (fibroblast-HLC^LT^) allowed expansion of the HLCs without significantly changing their gene expression profile and functionality. Intrasplenic injection of 10 × 10^6^ fibroblast-HLC^LT^ into FRG mice resulted in the survival of 33% of mice, with a RI of 0.3–4.2% in the surviving mice 9 weeks after transplantation. Transcript levels for *AFP,* but also *ALB* and *AAT,* in fibroblast-HLC^LT^ recovered from the mice, were lower than in the grafted cells, while *CYP3A4* expression was increased [118]. As an alternative, Du et al. overexpressed *HNF1α*, *HNF4α*, *hepatocyte nuclear factor 6* (*HNF6), activating transcription factor 5* (*ATF5), prospero homeobox protein 1* (*PROX1),* and *CCAAT/enhancer-binding protein*
*α* (*CEBPA).* These transdifferentiated fibroblasts (2 × 10^6^ cells) were grafted in *Tet-uPA Rag2^−/−^IL2rg^−/−^* mice yielding repopulation at efficiencies up to 30% and hALB levels up till 300 μg/mL, 7 weeks following transplantation [123].

#### 5.4.2. Transdifferentiation of Somatic Cells Towards Hepatocyte-Like Cells Via an Intermediate Progenitor Stage

An alternative approach is to dedifferentiate fibroblasts to an induced multipotent progenitor cell (iMPCs) using a combination of Yamanaka reprogramming factors (e.g., *OCT4*, *SOX2,* and *KLF4*) and then fate the cells to HLCs using hepatocyte supportive culture media. Using this strategy, Zhu et al. generated so-called iMPC-derived endoderm progenitor cells (EPCs) that could be extensively expanded, and subsequently fated to HLCs. iMPC-HLCs closely resembled fetal PHHs, expressing hepatocyte specific proteins such as HNF4α, AAT, and cytokeratin 18 (CK18), but low levels of CYP450 enzymes [120,122]. Intrasplenic injection of 1 × 10^6^ iMPC-HLCs into FRG mice, preconditioned with an adenovirus expressing *uPA*, resulted 9 months after transplantation in a RI of 2%, with detectable hALB secretion in mouse serum (levels ranging from 10–100 μg/mL). In line with the directly transdifferentiated fibroblast-HLCs from the Du et al. study [123], transcriptome studies of iMPC-HLCs, recovered following in vivo grafting, compared with non-transplanted iMPC-HLCs, and fresh and transplanted PHHs, demonstrated that iMPC-HLCs mature further in vivo [122].

Thus, direct or indirect TF-mediated transdifferentiation of human fibroblasts into HLCs yield cells with in vitro resemblance to fetal hepatocytes, which upon grafting in vivo can engraft in mouse liver, even if robust repopulation is not seen. However, long-term post-transplant follow-up has not been extensively documented. As maturation occurs in vivo, it remains possible that more robust repopulation would be seen at later time points, which is also what was observed when organoid derived progeny were grafted [42]. If higher degrees of repopulation could be attained, perhaps the indirect transdifferentiation approach would be more useful clinically, as the endodermal progenitor intermediate can be massively expanded. Nevertheless, the possibility of incorrect differentiation or even presence of some cells with (near) pluripotent characteristics might detract from this approach over directed transdifferentiation. A final thought is that it might be possible, as has been shown by Song et al., to introduce the most optimal complement of TFs that guides transdifferentiation of human hepatocytes in vitro into the mesenchymal compartment of the liver to directly transdifferentiate non-hepatic cells in hepatocytes, rather than performing the transdifferentiation in vitro [124].

### 5.5. Pluripotent Stem Cell Derived Hepatocytes

Pluripotent stem cells (PSCs) have been touted to be the ultimate cell population from which to create hepatocytes (and many other cells) for regenerative medicine purposes. PSC can self-renew without senescence, and differentiate into any cell type of the human body [125,126]. Human embryonic stem cells (hESCs) are derived from the inner cell mass (ICM) of the human blastocyst and resemble epiblast cells [127]. They were first culture-isolated in 1998 by Thomson et al. [128]. As already discussed above, in 2007, the Yamanaka team developed technology whereby human fibroblasts (and since then any nucleated cell) could be dedifferentiated into the (near) equivalent of hESCs, cells they termed hiPSCs [116,129], opening the possibility for personalized regenerative therapies derived from autologous PSCs. However, as transplantation of undifferentiated PSCs causes teratoma formation, the quest has been to develop robust protocols to generate terminally differentiated lineage specific progeny, including cells with hepatocyte characteristics.

#### 5.5.1. Differentiation of Pluripotent Stem Cells Towards Hepatocytes Via Embryoid Body Formation

In early studies, differentiation of PSC towards HLCs comprised the formation of embryoid bodies (EBs). In this approach, differentiation to definitive endoderm occurs spontaneously due to paracrine signals emanated from the different cell types within the EB, followed by further differentiation towards cells with hepatocyte-like features by means of cytokines and/or small molecules [5,130,131,132,133,134]. EB-derived HLCs had some characteristics of hepatocytes, but resembled fetal, not adult, PHHs, as they were highly AFP^+^ and have poor CYP450 functional activity or urea metabolism [130,131]. Moreover, HLCs were quite heterogenous, possibly caused by the non-directed initial EB step.

Duan et al. grafted 5 × 10^5^ EB-derived ESC-HLCs directly in 2 different sites in the liver of NOD-SCID mice [132]. Small colonies of hALB and hAAT^+^ cells were found 3 weeks after transplantation, and low levels (10–40 ng/mL) hALB could be measured in the serum. Moreover, serum hALB levels decreased when measured beyond day 14 after transplantation, suggesting loss of grafted cells. This loss of grafted cells over time was also observed by Woo et al. [5]. In another study, Basma et al., found teratoma formation in the liver and spleen following transplantation of 1 × 10^6^ EB-derived hESC-HLCs in hepatectomized NOD-SCID mice [130]. To circumvent this problem, they isolated the 18%–26% fraction of asialoglycoprotein receptor (ASGPR) positive, presumed HLCs, cells from the EB-derived mixed cell population. Intrasplenic injection of 1–2 × 10^5^ ASGPR enriched HLCs into uPA-SCID mice resulted in small clusters of hCK18^+^ cells in the mouse liver 75 days post transplantation. Similar results were seen when the ASGPR enriched HLCs were grafted in hepatectomized FK506 Nagase analbumic rats. However, only low levels of hALB were detected. This is consistent with the notion that EB-derived HLCs are fetal-like, and do not yet resemble adult PHHs. Moreover, although the ASGPR enriched HLCs no longer caused teratoma formation, well-differentiated adenocarcinomas were found in the peritoneum of the rats, indicating that tumor formation remains a risk when using EB-derived HLCs and that a more directed and better controlled differentiation of PSCs towards hepatocytes is needed to prevent tumor formation.

#### 5.5.2. Directed Differentiation of Pluripotent Stem Cells Towards Hepatocytes

Numerous protocols have been developed to directly differentiate HLCs from hESCs/hiPSCs. In general, they depend on the step-wise addition of growth factors, cytokines and/or small molecules to cells cultured on plastic dishes, to mimic different steps in development [20,135,136,137,138,139]. In addition, as is also described for the fibroblast transdifferentiation studies, hepatic TFs have been overexpressed during PSC differentiation to enhance directed differentiation [140,141,142], cultures have been treated with miRNA inhibitors [143], or suspension and 3D spheroid cultures of HLCs have been established to enhance HLC maturation [144,145,146]. Although these directed differentiation protocols result in less heterogeneous progeny, even the most ‘mature’ hPSC-HLCs still resemble fetal hepatocytes as has been shown by transcriptome, proteome, and functional studies and this more or less irrespective of the differentiation process used [147,148,149,150]. However, the PSC-HLCs do express certain mature markers and therefore do not have a complete fetal phenotype [146].

A number of groups tested the repopulation ability of PSC progeny differentiated using a directed approach to the endoderm stage or a hepatic progenitor/hepatoblast stage. Liu et al. transplanted 0.1 to 2 × 10^6^ PSC derived DE cells, hepatoblasts (HBs), or final differentiated HLCs via the tail vein in *NOD/Lt-SCID/IL2rg^−/−^* mice, pretreated with dimethylnitrosamine. Eight weeks after transplantation, CYP2E1 and hALB^+^ clusters could be detected in the mouse livers (RI of DE cells: 4–10%; RI of HBs and HLCs: 2–6%). Low levels of hALB were also detected in the mouse. In contrast to the studies described above injecting EB-derived HLCs, Liu et al. found persistent presence of human cells for more than 7 months, and no tumor formation was detected [137]. The repopulation efficiency could be increased to 35%, when the grafted cell dose was increased to 7 × 10^6^ PSC-DE cells, demonstrating that engraftment and/or repopulation of PSC-HLCs, although possible, is significantly less efficient than that of PHHs [137]. Yan et al. grafted not fully differentiated PSC-hepatic progenitors (HPCs; 2–5 × 10^6^ cells) by direct injection in the liver of FRG mice, resulting in presence of ±25% human FAH and AFP^+^ cells 70 days after transplantation [138]. 

These studies demonstrate that hepatocyte progenitors (DE cells or HPCs) generated from PSCs can engraft and repopulate rodent livers, but with significantly lower efficiency than adult PHHs, and likely more consistent with the engraftment/repopulation observed after fetal hepatocyte transplantation.

Many more studies have evaluated the engraftment and repopulation ability of terminally differentiated hPSC-HLCs. For instance, Asgari et al. grafted 1 × 10^6^ PKH67-labeled HLCs into CCl_4_ treated Swiss mice by tail vein injection [136]. Five weeks after transplantation, all transplanted animals survived, while only 3 out of 5 non-transplanted animals survived. However, only few PKH67-labeled HLCs were detected in the mouse liver (RI of 2%). By contrast, Carpentier et al. reported RIs up to 20%, 100 days following grafting HLCs intrasplenically into MUP-uPa-SCID/Bg mice, which have the *uPA* gene under control of the major urinary protein (MUP) promoter, even if the degree of repopulation varied between 1 and 20%.

As most studies reported relatively low levels of repopulation, investigators tried strategies that might enhance engraftment by among others, modifying immune rejection, enhancing liver damage, and/or enhancing donor hepatocyte proliferation. Nagamoto et al. transduced PSC-HLCs with an adenoviral vector overexpressing a hyperactive *Bcl-xL* gene (ad-FNK) to inhibit donor cell apoptosis [142]. Grafting of 1 × 10^6^ Ad-FNK-transduced PSC-HLCs in the spleen of uPA-SCID mice resulted in a RI of 20% 4 weeks after transplantation, which was significantly higher than when untransduced HLCs were grafted. However, engraftment of PSC-HLCs was also found into other organs. Therefore, the investigators also tested if transplantation of a genetically non-modified hPSC-HLC monolayer sheet directly into the liver of partially hepatectomized mice could overcome the spread of HLCs to other organs [141]. These studies showed that grafting a sheet of HLCs in the liver rather than infusion in the spleen not only prevented migration of HLCs in other organs, but also enhanced the levels of hALB in mouse serum and significantly reduced acute liver injury caused by administration of CCl_4_ compared to intrasplenically grafted mice. This might suggest that PSC-HLCs fail to properly home to the liver when infused via the portal vein (when transplanted in the spleen), but have the ability to engraft and expand when placed directly into the liver parenchyma. However, repopulation in the liver was only seen near the location where the sheet had been implanted, and not throughout the liver. An alternative approach was described by Yuan et al., who grafted 3 × 10^6^ iPSC-HLCs in the spleen of FRGS mice which were treated weekly with XMU-MP-1, an inhibitor of pro-apoptotic MST1/2, and JO2, a mouse CD95 antibody to continuously further kill endogenous murine hepatocytes, for the first 6 weeks after transplantation. This resulted in a progressive increase in hALB^+^ cells from 26% to 44% 20 weeks after transplantation [20]. Similar results were seen when the JO2 antibody was combined with a 5D5 antibody, an agonist of the c-Met receptor that promotes cell proliferation, which increased the repopulation from 20% to 42% 8 weeks after transplantation. hALB levels detected in the mouse serum were in the same range (1–2 mg/mL) as observed in animals transplanted with uncultured PHHs. No tumorigenesis or migration into other organs was seen [21].

Thus, PSC-HLCs generated via directed differentiation remain fetal hepatocyte-like, which is likely responsible for their poor engraftment and repopulation ability in the different rodent models tested. However, engraftment/repopulation can be increased if apoptosis of the HLCs is decreased by transduction with a hyperactive form of *Bcl-xL* [141,142], suggesting that immature HLCs may be more susceptible to cell death than mature PHHs. In addition, studies wherein c-Met is activated [20], known to cause enhanced proliferation of hepatocytes, also enhanced the engraftment/repopulation efficiency, suggesting that the proliferation inducing signals in the mouse hepatocyte niche, while sufficient to induce proliferation and repopulation of PHHs, are insufficient for HLC proliferation and repopulation; what is also believed to be the case for grafting fetal primary hepatocytes [45]. A similar mechanism may be operative when continuously more severe hepatocyte damage is caused by administration of JO2 antibody [20,21]. Finally, inability to home to the liver may also be a reason why HLCs poorly engraft/repopulate mouse livers, as implantation of HLC sheets in the liver also improved (albeit local) repopulation [141].

## 6. Discussion

Mature hepatocytes, isolated from livers that are not suitable as donors for whole organ transplantations remain the gold standard source of cells for liver repopulation. However, as they are a scarce commodity, researchers are actively looking for alternative cell sources for cell-based liver transplantations. If these alternative cell populations could be generated from the patient him/herself, they may also circumvent the need for immunosuppressive treatment. In this review, we highlight that currently, none of the described alternative cell populations have the same ability to engraft and repopulate the liver to a similar level as PHHs, although some of them (MSCs and AECs) have been proven to be successful in alleviating symptoms of certain liver diseases, even without repopulating the host liver. Although, engraftment of the alternative cell sources (e.g., 2D cultured hepatocytes, LPCs, liver organoids, and (trans)differentiated HLCs) can be observed in the mouse livers, they fail to repopulate the injured livers to the same extent as PHHs (see Table 2). This is not surprising, as fully mature PHHs have a significantly higher propensity to engraft and repopulate the liver, compared with for instance fetal hepatocytes [42,43,44,45]. When PHHs are expanded in vitro, they dedifferentiate (i.e., they undergo epithelial-to-mesenchymal transition and rapidly loose mature hepatic functions within 24–72 hours after culturing) and hence acquire an immature more fetal-like phenotype [151,152]. This is also true for all the alternative cell populations tested until now.

We believe that the failure of cells, other than mature PHHs, to robustly engraft and repopulate the host liver are two-fold (Figure 2): (1) immature or dedifferentiated hepatocytes cannot home efficiently into the liver microenvironment, and (2) inability of the engrafted cells to survive and proliferate in the host liver. These conclusions follow from the following experimental observations:

First, in strong contrast with PHHs, immature cells (be it fetal hepatocytes, dedifferentiated and expanded PHHs, LPCs, transdifferentiated “hepatocytes”, or PSC-HLCs) may not sufficiently recognize signals for invading and engrafting into the adult hepatic tissues when administered via the blood circulation. This can explain why engraftment efficiency drastically increases when these cells are directly introduced as a sheet of cells in the liver parenchyma [141].

This problem will only be solved once we understand the cellular and molecular processes underlying the poor homing and engraftment potential of the immature cells. It is believed that hepatocytes, when transplanted via the circulation, become entrapped in the liver sinusoids, through which they need to migrate to reach the liver parenchyma. When trapped in the sinusoid, this causes transient portal hypertension, and due to a transient ischemia-reperfusion injury, Kupffer cells become activated, induce endothelial activation and enhance vascular permeability. Hepatocytes then attach to the endothelium and migrate through the fenestrae to home into the parenchyma. Although this process is not very well understood, it is believed that interactions between grafted hepatocytes, Kupffer cells, liver sinusoidal endothelial cells, stellate cells, and the surrounding extracellular matrix all play a role in this homing and initial engraftment process [153,154,155]. The process is relatively inefficient, as even for the gold-standard adult liver derived PHHs, it is believed that only a small fraction of hepatocytes can home into the parenchyma. Some studies have been done to track whether transplanted cells engraft into the liver using, for instance, superparamagnetic iron oxide nanoparticles (SPIO) labeled hepatocytes. One drawback of this approach is that activated Kupffer cells and macrophages in and around the sinusoid lead to very fast scavenging of dying SPIO labeled hepatocytes, making interpretation by magnetic resonance imaging (MRI) of the presence of hepatocytes difficult [156]. As an alternative, Indium-111-labeled cells could be used, as has been done clinically [157]. Therefore, to devise methods that would enhance the homing and engraftment of alternative sources of hepatocytes, these processes will need to be better studied.

Using RNA sequencing and/or proteomics, defining the molecular make-up of for instance fetal and adult PHHs, different batches of PHHs with differing engraftment potential, or possibly even better, genetically identical, early and late passage ex vivo expanded PHHs (both in 2D and 3D) should also aid in deciphering why specific populations have a reduced engraftment and repopulation potential. The observation that some human and mouse HLCs may be more capable of engrafting the liver, when they are harvested from the primary grafted animal, and used to engraft into secondary mice, suggests that once cells have adapted to an adult in vivo liver microenvironment, engraftment capacities are boosted. Further expanding on the molecular studies that evaluated the re-isolated hepatocytes with single cell/single nuclei RNA sequencing studies in comparison with the initially grafted cells might therefore also provide insights in what molecular changes are occurring in the cells when placed in the in vivo liver environment, and might provide clues on how to enhance homing and engraftment [118,122].

Except for generating cells that are better adapted to the in vivo hepatic environment, this environment itself has a big influence on the engraftment efficiency. A strong correlation exists between the level of liver damage and engraftment/repopulation efficiencies; however, inducing more extensive liver damage to enhance homing is not clinically translatable. Moreover, the more severe the injury, the greater the inflammatory response, which then may lead to excess scavenging by Kupffer cells/macrophages of the grafted cells. It is interesting to note that studies have demonstrated that co-transplantation of hepatocytes and MSCs increases the number of donor hepatocyte clusters in the liver, which was attributed to the fact that the cells home better to the liver [158] by enhancing attachment of the hepatocytes. In addition, the strong immunomodulatory/anti-inflammatory effects of MSCs may aid in the initial survival of donor hepatocytes in the inflamed liver niche environment [159]. Therefore, it might be interesting to test if co-culture or co-infusion of MSCs (and/or AECs), that themselves cannot repopulate the liver parenchyma, with alternative sources of hepatocytes can enhance the latter’s homing and initial survival.

The second important issue to address is how to enhance proliferation (and survival) of engrafted cells to obtain repopulation levels that are clinically relevant. Indeed, introducing an anti-apoptosis gene in HLCs improved their survival in vivo [142], and providing chemical compounds that support proliferation also appeared to enhance repopulation of PSC-HLCs [20,21]. Although interfering with cell death processes and enhancing cell proliferation hold intrinsic risks for enhancing tumor formation, we believe that devising strategies that would increase proliferation may be an interesting avenue to improve liver repopulation by immature hepatocytes, the more that some of these compounds have also been suggested to increase regeneration potential of endogenous hepatocytes [160]. However, if the maturation state of the alternative cell sources could be improved, these measures may not be required. Furthermore, we also have to keep in mind that that the risk of tumorigenesis remains due to incomplete differentiation of the transdifferentiated hepatocytes or PSC-HLCs. Better differentiation protocols will therefore also be needed to overcome this problem. 

We conclude that none of the alternative sources of “hepatocytes” for liver repopulation currently can robustly engraft and repopulate the liver. Comparing cells derived from the liver (2D expanded hepatocytes, liver progenitor cells, and organoids) with cells derived from tissues other than liver (transdifferentiated hepatocytes or PSC-derived hepatocytes), no very clear differences exist in their ability to home, engraft, and proliferate in the host liver; and some differences in repopulation ability reported appear to be mainly due to differences in animal models used, and the inclusion or not of additional immunomodulating, anti-apoptosis, and proliferation inducing modifications in the engraftment studies. 

From a clinical perspective, the optimal cell source would be cells that can be obtained from easily accessible and autologous sources, that can be easily expanded to multiple billions of cells, and that do not hold the risk for tumor formation. iPSC-derived HLCs best fulfill the first two criteria, whereas expanded cells (in 2D or as organoids) likely hold less chance for tumor formation. Nevertheless, neither of these cell technologies are currently fully mature, and hence underperform significantly in regards to engraftment and repopulation compared with PHHs. However, progress over the last 5–10 years has been remarkable. Ongoing and future studies are now aimed at understanding how to create more mature progeny from in vitro hepatocyte sources. Studies should also identify means to improve homing/engraftment and subsequent proliferation and repopulation by the less mature cells we can currently generate, and/or to enhance the receptiveness of the liver microenvironment for hepatocyte engraftment. Such insights will then pave the way for starting to test regenerative strategies starting from cultured hepatocyte progeny.

## Figures and Tables

**Figure 1 cells-09-00566-f001:**
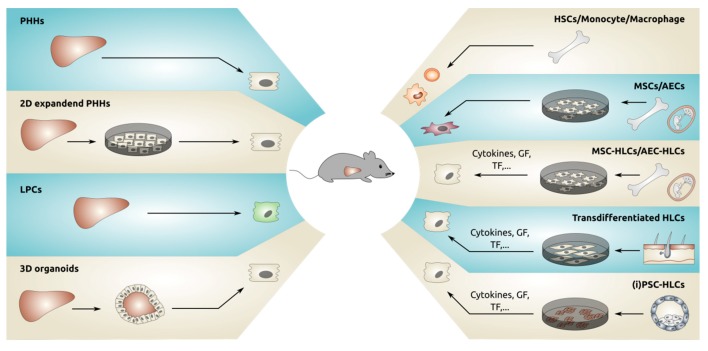
Different sources of cells that have been transplanted in mouse livers. GF = growth factors, TF = transcription factors, PHHs = primary human hepatocytes, LPCs = liver progenitor cells, HSCs = hematopoietic stem cells, MSC = mesenchymal stromal cells, AEC = amniotic epithelial cells, iPSC = induced pluripotent stem cells, HLCs = hepatocyte-like cells.

**Figure 2 cells-09-00566-f002:**
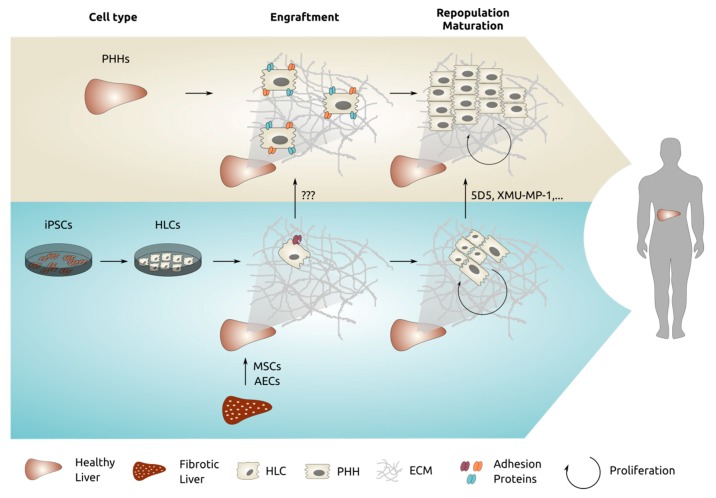
Overview of the processes we believe are required to ensure proper engraftment and repopulation of PHHs into human liver. In blue, the reduced engraftment and repopulation efficiency of iPSC-HLCs is shown, and some strategies to potentially overcome these drawbacks are indicated.

**Table 1 cells-09-00566-t001:** Definitions of different effects exerted by transplanted cell populations on the liver, used in this review.

Phenomenon	Definition
**Liver engraftment**	Engraftment is the initial incorporation of transplanted cells into the liver tissue, whether or not the transplanted cells proliferate to replace a significant proportion of the liver mass.
**Liver repopulation**	Repopulation occurs following engraftment and is the result of extensive proliferation of transplanted cells that have hepatocyte functions in the host liver. In this review, presence of at least 5–10% of donor cells in the host liver 2 months post transplantation is considered ‘successful liver repopulation’. This cut-off was defined based on the observation that in many instances, 5–10% of the hepatocytes must be replaced to obtain a therapeutic effect of the transplanted cells [7].
**Serial transplantation**	Serial transplantation is defined as the isolation of donor hepatocytes from a primary host and the engraftment of these isolated donor hepatocytes into a secondary host, which is possible as hepatocytes can undergo a number of cell divisions [8].
**Cell fusion**	After transplantation of cells into the liver (best illustrated following transplantation of monocytes), donor cells can fuse with host cells, thereby generating (either mono- or bi-nucleated) polyploid cells. Genes expressed from the donor genome could then compensate for low or absent gene expression from the host genome (e.g., due to inborn genetic disorders). Such events could easily be misinterpreted as transdifferentiation.

**Table 2 cells-09-00566-t002:** General overview of hepatocyte transplantations in pre-clinical animal models as described in the main text and grouped per cell type. Other differences, like differences in animal models, route of cell administration, and differentiation protocol, were not considered. Late = 2 months or more, Early = less than 2 months, ND = no data available.

	Characteristics of Cells to Transplant	Repopulation Index (%)		hAlb Blood Levels		Mechanism	Serial Transplantation	References
		*Early*	*Late*	*Early*	*Late*			
PHHs	Alb^+^ AFP^−^	10%–50%	20%–95%	1–5 mg/mL	5–15 mg/mL	Functional integration in liver parenchyma	Yes	[13,17,26,31,33,34]
2D expanded PHHs	Passage < 4	ND	50%–90%	<500 μg/mL	2–10 mg/mL	Functional integration in liver parenchyma	ND	[49,50]
	Passage > 6	ND	1%–40%	<10 μg/mL	1–5 mg/mL			
LPCs	Several markers have been used, but mostly fetal cells	<5%	0%–10%	0–1 mg/mL	<100 μg/mL	Functional integration in liver parenchyma	Yes	[44,45,51]
3D organoids	Alb^+^ AFP^+^	ND	ND	<100 μg/mL	ND	Functional integration in liver parenchyma	ND	[42]
HCSs/Monocytes/ Macrophages	CD34^+^/CD14^+^	0%	10%–30%	ND	ND	Fusion with host hepatocytes	ND	[3,60,61]
MSCs	Several markers have been used	0%–10%	0%–50%	0–1 mg/mL	0–2 mg/mL	Immunomodulation, Paracrine effects, …	ND	[9,69,81]
MSC-HLCs	Alb^+^ AFP^+^	1%–5%	ND	0–10 ng/mL	ND	Immunomodulation, Paracrine effects, …	ND	[86,89,97,98]
AECs/AEC-HLCs	Several markers have been used/Alb^+^, AFP^+^	0.1%–3.5%	0.1%–5%	ND	ND	Immunomodulation, Paracrine effects, …	ND	[111,113]
Transdifferentiated HLCs	Alb^+^ AFP^+/-^	1%–30%	~2%	0–300 μg/mL	<100 μg/mL	Functional integration in liver parenchyma	ND	[118,122,123]
(i)PSC-HLCs	Alb^+^ AFP^+^	1%–20 %	1%–45%	0–2 mg/mL	0–2 mg/mL	Functional integration in liver parenchyma	ND	[20,21,137,141,142]

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
