# Peer review of "Alternative Cell Sources for Liver Parenchyma Repopulation: Where Do We Stand?"

_cells, 2020, doi:10.3390/cells9030566_

Round 1

Reviewer 1 Report

The manuscript submitted by Tricot and colleagues aims to offer an overview on alternative therapy to threat acute or congenital liver disorders. Such kind of review article is not new. Review articles like this come in addition to a large plethora of review articles describing hepatocyte transplants and stem cell-derived hepatocyte-like cells as adjunct treatment for acute and congenital disorders. 27 years of clinical hepatocyte infusion have generated more review articles than real original research or clinical reports.

Nevertheless, the present manuscript is clear and well written. Figure are original and informative. However, two major limits are strongly affecting the current review: the lack of criticism and recent evidence in support of an immature phenotype generated by any kind of stem cell-derived hepatocyte-like cells generated so far, with any in vitro technology (see Zabulica et al., Stem Cell Development 2019 for recent evidence of fetal maturation). Second limitation, the lack of description for perinatal stem cells, in particular human amniotic epithelial stem cells, which have been largely described during the past 10 years to correct amino acid and neurotransmitter imbalances in relevant animal models of liver diseases, at a level never reached by any other stem cells before. 

Several other progenitor cells, including mesenchymal stromal cells, which have been proved to survive for very short time upon transplantation, or hematopoietic stem cells, which have been elegantly shown to support hepatic function only when fusion between recipient hepatocyte and donor cells occurred, have so far generated limited clinical benefits and resulted in expensive but inconclusive clinical trial for a long-term correction of human disorders.

The authors should probably be a little bit more critic and describe limitations and benefits resulting from all the alternative treatments proposed and described (including immature hepatic phenotype and tumorigenic risk associated with third party donation). 

Spanning from multiple stem cell sources (including liver stem cells which have never been proved or described in human), to recent organoidal in vitro formations and embryonic bodies requires a very clear structure and critical description for all such approaches, with particular attention to preclinical evidence of success. The use of relevant experimental models is of outstanding importance, to examine the efficacy and safety of the product.

We encourage the authors to revise and critically editing their manuscript, in comparison with precedent similar manuscript published every where in scientific literature (including CELLS) before resubmitting for publication. 

Author Response

We would like to thank the reviewer for the suggestions made.

We fully agree with the reviewer that the hepatocyte-like cells (HLCs), that are generated in vitro have an immature phenotype, which we discussed in our initial draft in detail in the main text as well as in the discussion part. However, as suggested by the reviewer, we also included a section incorporating the notion that although HLCs are immature, they may be somewhat less so than true fetal hepatoblasts (as demonstrated in the Zabulica et al., paper) into the main text (lines 501-511). We also included a section on the amniotic epithelial cells (lines 315-316 and 338-343). We also shortly discussed the risk of tumorigenesis when using immature stem cell derived hepatocyte-like cells for liver transplantation.

Reviewer 2 Report

This is a very broad overview of the field. Although none of the aspects has been extensively covered, it can be of interest to a general readership

Author Response

We thank the reviewer for reading the manuscript.

Reviewer 3 Report

The current manuscript from Tine Tricot et al. entitled ‘Hepatocytze Transplantation from Alternative Cell Sources: Where Do We Stand Now?’ gives a profound overview of the used animal models and the current used cell types in the background of liver regeneration. The Manuscript is written nicely and includes all cell types and techniques which were used so far in this field. This reviewer has only one minor note.

In line 464 (p 11) and 549 (p 13) the amount of the detected human albumin in the host blood sera was forgotten.

Therefore this reviewer pronounce this manuscript for publication.

Author Response

We thank the reviewer for reading the manuscript and pointing out missing details concerning the concentrations of albumin detected in the host blood. We have included this data on line 474 and 566 of the manuscript.

Reviewer 4 Report

In this review paper, the authors summarized the background and current studies of hepatocyte transplantation. First, they introduced commonly used animal models and grafting of primary human hepatocytes. Second, they discussed the approaches to obtain large number of hepatocytes, including methods to stimulate more hepatocytes from primary liver tissue and to do so from other tissues. Finally, the authors wrapped up the review and provided some ideas regarding the field. Overall speaking, this is a detailed review and can be beneficial to researchers in related fields.

Line 204, is it 3-D cultured hepatocytes?

Author Response

We thank the reviewer for reading the manuscript and the constructive comments. We have corrected the subtitle to 2D cultured hepatocytes (line 204).

Round 2

Reviewer 1 Report

R1 version of the review manuscript submitted by Tricot and coll. contains limited corrections and clarifications. The manuscript is still suffering from inaccuracies and incomplete information, in example:

The definition of serial transplantation is partial and inaccurate. Hepatocyte transplantation has been proposed as alternative treatment since, in contrast to liver transplant, it can be performed repeatedly, without removing the native organ. Repeated infusion is not only feasible but largely recommended to reach the clinical dose for correction in congenital liver disorders (up to 15 billion.

Moreover, cell fusion (like in BM-derived MSC Transplantations) is not occurring between cell nuclei. It has been described that donor and host cell fuse without combining genetic material. Polyploidy in liver cells is a common phenomenon, but insufficient to correct genetic disorders

Interestingly, the authors did not mention differences between freshly isolated and criogenic human hepatocytes (which have been proved significantly different both in term of survival and function). In Figure 1, the authors mentioned primary human hepatocyte expanded in vitro in 2D conditions up to 6 passages. Such assumption should be motivated and supported.

The major limitation in the current manuscript stands on the clear definition and explanation of different cell population extensively studied and transplanted to correct acute and congenital liver disorders (with different results).

Hepatic adult progenitors have been largely described in murine models, however, so far, no clear evidence has been reported to support their significance in human liver injury repair. However, an elegant and accurate description of bipotential cell progenitor in chronically injured liver has been offered by Grompe’s group (Tarlow et al., Cell Stem Cell 2014). In such seminal manuscript, the authors provided accurate experimental evidences in support of metaplasia rather that stem cell differentiation. In such important (but unfortunately uncited) manuscript, the authors conclude that human and mouse hepatocytes can undergo reversible metaplasia, generating both cholangiocyte and hepatocyte functional mature cells.

LPC described by Sokal et al. (mentioned in the current manuscript) has been described as liver-resident stromal cells. Thus, they should be included in the MSC section.

On the contrary, the cells that have been clearly and accurately described NOT to be MSC-like cells are the amniotic epithelial cells. Amnion membrane, isolated from human placentae, contains two distinct cell population: amnion-derived MSC and amnion epithelial cells. Amnion epithelial cells are clearly epithelial by definition and have been largely described negative for almost all the markers commonly associate with MSC. The capacity to differentiate into all three germ layer cells (including mesoderm and MSC-like) ascribed to sole AEC has often let these cells to be misled with MSC. We encourage the authors not to commit such common mistake. Marongiu et al. (Hepatology 2011), Manuelpillai et al. (Cell Transpl 2010), Skvorak et al. (Hepatology 2014), Liu et al. (Stem Cell Res & Ther 2018) clearly described the potential of such epithelial multipotent cells in treating acute or congenital liver disorders. And the level of hepatic maturation and long-term survival in the host recipient has been proved superior and safe compared to any other stem or progenitor cells ever measured before.

Page 9 (line 342). The statement that “low engraftment efficiency in human AECs into mouse liver have been observed (0.1-1%)” is actually incorrect. In Skvorak et al. (Hepatology 2014), the amount of human DNA (AEC donor) has been measured equal to. 4% (approx. 80% efficiency in cell engraftment)

Another comment at page 9 (line 354-356) “we believe that MSC are not good candidates for liver repopulation, they might hold potential for grafting together with (stem) cells that have the potential to regenerate the liver” is actually unclear. Which stem or third part donor cells? Please clarify

The animal models section is quite superficial and sometime incomplete. PH and toxicological models are hastily described, in comparison with transgenetic models. FRG model is correctly and largely described, without unfortunately mentioning that apart from the previous version described by Azuma et al. in 2007, lately same model has been improved and largely commercialized (by Yecuris) as FRGN model (Wilson et al. 2014) to enhance hep engraftment and repopulation

The statement at page 4 (line 122) that “it is also possible to enhance the proliferation efficiency of the grafted hepatocytes by a treatment” with different molecules is quite pretentious. Unfortunately, several approaches have been implied to enhance donor cell engraftment, as well as encouraging proliferating in engrafted cells. Unfortunately, apart from a selective growth advantage over resident hepatocytes played by genetic competence (as clearly proved in FRG mice), nothing has significantly and clearly proved efficacy so far. However, a mention on such molecular approaches is correct but the authors should avoid over- interpretations.

At line 509, the authors correctly stated that hepatocyte-like cells generated by any stem or stromal cells “do not have a complete fetal phenotype”. Actually, such affirmation should specify that, with current technology, the best level of maturation commonly achieved is resembling a fetal hepatocyte phenotype. Such insufficient level of functionality has been proved in any kind of progenitor cells.

Later in the text, at page 14, line 585, the authors state “In this review, we highlight that currently, none of (described, ndr) alternative cell populations have the same ability to engraft and repopulate the liver as PHHs”. Such assertion may actually consider true but it needs to be elaborated a little bit more to not lead the reader to misinterpretations.

Similarly, “When PHHs are expanded in vitro, they de-differentiate and acquire an 589 immature more fetal-like phenotype” requires a better explanation, due to the limited/null capacity of primary human hepatocyte to grow in in vitro conditions. Epithelial-to-mesenchymal transition or dedifferentiation and loss of hepatic functions have been largely described in >7 days in vitro experiments. Please elaborate.

Regarding the human hepatocyte transplants and engraftment, the authors mentioned “studies to track homing of hepatocytes into the liver, using for instance superparamagnetic iron oxide nanoparticles (SPIO)”, while the evidence that transplanted cells do not readily pass through the liver to extrahepatic sites was provided when human hepatocytes were labeled with Indium-111 oxyquinolone and infused in a 5-year-old child with urea cycle defect (Bohnen et al. Clin Nucl Med 2000)

In conclusion, it’s reviewer’s opinion that the state-of-the-art manuscript would sound better and easy to read if shorten version is uploaded. A critical editing/shortening in several parts of the manuscript may largely benefit the authors’ scope. The authors should focus on one scope only, i.e., alternative cell sources to hepatocytes, without navigate within the complicate field of hepatocyte transplant, animal models or in vitro techniques for cell expansion, which may confuse the reader. A critical review of the literature and correction in the nature of different cell population currently proposed for different liver diseases (i.e., pluripotent vs multipotent cells) would be sufficient and of value for a manuscript like the one proposed by Tricot and coll.

Minor details:

Asialo-GM1 should be clearly stated at first Tricot and coll. is particularly long and unfortunately superficial in several parts. Table 2 appears to be over-simplified and not considering variability in animal model and donor cells

Author Response

R1 version of the review manuscript submitted by Tricot and coll. contains limited corrections and clarifications. The manuscript is still suffering from inaccuracies and incomplete information, in example:

The definition of serial transplantation is partial and inaccurate. Hepatocyte transplantation has been proposed as alternative treatment since, in contrast to liver transplant, it can be performed repeatedly, without removing the native organ. Repeated infusion is not only feasible but largely recommended to reach the clinical dose for correction in congenital liver disorders (up to 15 billion.

There might to be some confusion between serial transplantation (as defined in our review in Table 1), where transplanted cells from a primary recipient are transplanted in a secondary recipient, versus multiple transplantation, which is the repeated administration of hepatocytes to the same animal/patient (as mentioned by reviewer 1). Serial transplantation as we define it (similar to the definition used in hematopoietic stem cell research), can be used as a measure for the maturity of hepatocytes after transplantation in the first recipient.

Moreover, cell fusion (like in BM-derived MSC Transplantations) is not occurring between cell nuclei. It has been described that donor and host cell fuse without combining genetic material. Polyploidy in liver cells is a common phenomenon, but insufficient to correct genetic disorders.

We clarified the definition of cell fusion in Table 1.

Interestingly, the authors did not mention differences between freshly isolated and criogenic human hepatocytes (which have been proved significantly different both in term of survival and function). In Figure 1, the authors mentioned primary human hepatocyte expanded in vitro in 2D conditions up to 6 passages. Such assumption should be motivated and supported.

We agree with the reviewer that cryopreserved PHHs often have a poorer cell viability compared to freshly isolated PHHs, which has to be considered upon transplantation. Furthermore, functionality and plating ability are significantly poorer compared to fresh PHHs (Ostrowska et al., Cell and Tissue banking, 2000; Terry et al., Cell Transplantation, 2005; Iansante et al., Pedriatric Research, 2018). However, successful transplantations have been done with cryopreserved PHHs (example: Tateno et al., American Journal of Pathology, 2004) and it should be noted that the cryopreserved PHHs might be more user-friendly in a preclinical setting, than transplantation of fresh isolated PHHs. We added this to the text (Line 187-192).

We specified in the main text that the ‘6 passage of 2D cultured hepatocytes’ are a common factor seen in the references we cited, upon which the repopulation potential in mouse livers decreases (Line 235-236). However, to prevent confusion, and to present Figure 1 as a general overview figure, we removed such details from the figure.

The major limitation in the current manuscript stands on the clear definition and explanation of different cell population extensively studied and transplanted to correct acute and congenital liver disorders (with different results).

Hepatic adult progenitors have been largely described in murine models, however, so far, no clear evidence has been reported to support their significance in human liver injury repair. However, an elegant and accurate description of bipotential cell progenitor in chronically injured liver has been offered by Grompe’s group (Tarlow et al., Cell Stem Cell 2014). In such seminal manuscript, the authors provided accurate experimental evidences in support of metaplasia rather that stem cell differentiation. In such important (but unfortunately uncited) manuscript, the authors conclude that human and mouse hepatocytes can undergo reversible metaplasia, generating both cholangiocyte and hepatocyte functional mature cells.

We agree with reviewer 1 that the study by Tarlow et al., provided elegant evidence of liver repair mechanisms after injury. However, in our review we did not want to focus on natural liver repair. In the introduction, we shortly touched upon the fact that many repair strategies can be used by the liver, and referred to a very recent review focusing on this topic (in which the study by Tarlow et al., is cited). We then proceeded by saying how liver failure occurs if the natural repair responses are impaired, and concluded that in such cases liver transplants, or alternatively, hepatocyte transplantation is required. In this way, we made the transition to the actual topic of the review: finding useful cell sources for hepatocyte transplantations.

LPC described by Sokal et al. (mentioned in the current manuscript) has been described as liver-resident stromal cells. Thus, they should be included in the MSC section.

We agree with the comment of reviewer 1 about the mesenchymal-like phenotype of these cells. However, as the cells have also been described as progenitor cells, we choose to mention them in this section. However, we added the original terminology as used by Sokal et al., and changed ‘liver progenitor cells’ to ‘adult mesenchymal-like liver progenitor cells’ (Line 247). Furthermore, in the section about the progenitor cells, we stressed the fact that both fetal and adult progenitor cells have been used (Line 241). However, if the reviewer still prefers that we include this section into the MSC section, we are willing to do so.

On the contrary, the cells that have been clearly and accurately described NOT to be MSC-like cells are the amniotic epithelial cells. Amnion membrane, isolated from human placentae, contains two distinct cell population: amnion-derived MSC and amnion epithelial cells. Amnion epithelial cells are clearly epithelial by definition and have been largely described negative for almost all the markers commonly associate with MSC. The capacity to differentiate into all three germ layer cells (including mesoderm and MSC-like) ascribed to sole AEC has often let these cells to be misled with MSC. We encourage the authors not to commit such common mistake. Marongiu et al. (Hepatology 2011), Manuelpillai et al. (Cell Transpl 2010), Skvorak et al. (Hepatology 2014), Liu et al. (Stem Cell Res & Ther 2018) clearly described the potential of such epithelial multipotent cells in treating acute or congenital liver disorders. And the level of hepatic maturation and long-term survival in the host recipient has been proved superior and safe compared to any other stem or progenitor cells ever measured before.

We would like to thank reviewer 1 to point out our incomplete description of amniotic epithelial and amniotic mesenchymal stem cells. Accordingly, we added a subsection about AECs and their use in liver transplantation studies, and we discussed their engraftment and repopulation potential (Line 391-411).

Page 9 (line 342). The statement that “low engraftment efficiency in human AECs into mouse liver have been observed (0.1-1%)” is actually incorrect. In Skvorak et al. (Hepatology 2014), the amount of human DNA (AEC donor) has been measured equal to. 4% (approx. 80% efficiency in cell engraftment).

We would like to thank the reviewer to point out this reference, and we accordingly included it in the main text. However, the engraftment efficiencies were still much lower (just like with all the other alternative cell sources) than what is seen with PHHs (Line 404).

Another comment at page 9 (line 354-356) “we believe that MSC are not good candidates for liver repopulation, they might hold potential for grafting together with (stem) cells that have the potential to regenerate the liver” is actually unclear. Which stem or third part donor cells? Please clarify.

Following reviewer 1’s comment, we rephrased our conclusion in this section (Line 354-358).

The animal models section is quite superficial and sometime incomplete. PH and toxicological models are hastily described, in comparison with transgenetic models. FRG model is correctly and largely described, without unfortunately mentioning that apart from the previous version described by Azuma et al. in 2007, lately same model has been improved and largely commercialized (by Yecuris) as FRGN model (Wilson et al. 2014) to enhance hep engraftment and repopulation.

We added the latest new FRGN model to the manuscript (Line 105-108 and 180).

The statement at page 4 (line 122) that “it is also possible to enhance the proliferation efficiency of the grafted hepatocytes by a treatment” with different molecules is quite pretentious. Unfortunately, several approaches have been implied to enhance donor cell engraftment, as well as encouraging proliferating in engrafted cells. Unfortunately, apart from a selective growth advantage over resident hepatocytes played by genetic competence (as clearly proved in FRG mice), nothing has significantly and clearly proved efficacy so far.  However, a mention on such molecular approaches is correct but the authors should avoid over- interpretations.

We agree with the reviewer that we needed to nuance our statement and therefore changed the sentence to ‘might be possible’ and rephrased the sentence. Further, we also included citations of Yuan et al. that used these compounds to enhance the proliferation potential of engrafted hepatocytes in the livers of FRGS mice (Line 125-130).

At line 509, the authors correctly stated that hepatocyte-like cells generated by any stem or stromal cells “do not have a complete fetal phenotype”. Actually, such affirmation should specify that, with current technology, the best level of maturation commonly achieved is resembling a fetal hepatocyte phenotype. Such insufficient level of functionality has been proved in any kind of progenitor cells.

We added this to the text (Line 512-515).

Later in the text, at page 14, line 585, the authors state “In this review, we highlight that currently, none of (described, ndr) alternative cell populations have the same ability to engraft and repopulate the liver as PHHs”. Such assertion may actually consider true but it needs to be elaborated a little bit more to not lead the reader to misinterpretations.

We stated in the text that the described cell sources do have the potential to engraft, however not the potential to repopulate mouse livers to a similar extent as PHHs (Line 585-591). We hope this clarifies this statement. Furthermore, the reader can find a simplified overview of the repopulation potential of each cell type tested in Table 2 for additional information. Finally, as we also described in the previous manuscript version, we state that this failure to repopulate most likely is caused by the immature fetal-like state of the alternative hepatocyte sources.

Similarly, “When PHHs are expanded in vitro, they de-differentiate and acquire an immature more fetal-like phenotype” requires a better explanation, due to the limited/null capacity of primary human hepatocyte to grow in in vitro conditions. Epithelial-to-mesenchymal transition or dedifferentiation and loss of hepatic functions have been largely described in >7 days in vitro experiments. Please elaborate.

We agree with the reviewer and have added extra explanation about dedifferentiation as was suggested by the reviewer. Furthermore, 2 citations (Godoy et al. 2016 and Heslop et al. 2016) of papers that investigated this phenomenon extensively were added (Line 593-595).

Regarding the human hepatocyte transplants and engraftment, the authors mentioned “studies to track homing of hepatocytes into the liver, using for instance superparamagnetic iron oxide nanoparticles (SPIO)”, while the evidence that transplanted cells do not readily pass through the liver to extrahepatic sites was provided when human hepatocytes were labeled with Indium-111 oxyquinolone and infused in a 5-year-old child with urea cycle defect (Bohnen et al. Clin Nucl Med 2000).

There might be some confusion about what we describe as ‘homing’. We define it as the engraftment of the cells into the liver parenchyma and them persisting there. We agree with the reviewer that indeed, as shown in Bohnen et al. Clin Nucl Med, 2000, transplanted cells will not go through the liver to extrahepatic sites. However, not all the transplanted cells are able to engraft and thus fail to ‘home’ in the liver, upon which they do not survive. To clarify this, we adjusted the sentence in the main text. However, as correctly indicated by the reviewer, Indium labelled cells could also be used to track initial trapping and homing, without the confounding issue of long-term Indium positivity of macrophages, resulting from taking up the grafted cells. This has now been added to the text with the reference suggested by the reviewer (Line 617-618+621-622).

In conclusion, it’s reviewer’s opinion that the state-of-the-art manuscript would sound better and easy to read if shorten version is uploaded. A critical editing/shortening in several parts of the manuscript may largely benefit the authors’ scope. The authors should focus on one scope only, i.e., alternative cell sources to hepatocytes, without navigate within the complicate field of hepatocyte transplant, animal models or in vitro techniques for cell expansion, which may confuse the reader. A critical review of the literature and correction in the nature of different cell population currently proposed for different liver diseases (i.e., pluripotent vs multipotent cells) would be sufficient and of value for a manuscript like the one proposed by Tricot and coll.

We believe that, to understand all the transplantations performed in the mouse models, a background about the mouse models used, should be given to the reader so the reader does not have to do extensive extra research to understand each model used, and similarly to understand how all differences in in vitro cultures might influence transplantation capacity of the cells. However, we have shortened sections 4 and 5.

Minor details:

Asialo-GM1 should be clearly stated. at first Tricot and coll. is particularly long and unfortunately superficial in several parts. Table 2 appears to be over-simplified and not considering variability in animal model and donor cells.

Anti-asialo-GM1 is explained to be a natural killer cell inhibitor on line 122-123.

Table 2 is meant to be a general overview based on the different donor cells, not on the animal models. The reader can find more details about the differences between the mouse models in the main text. We included it as a simple overview for the reader and to clearly show that currently, no alternative cell population reaches the same level of liver repopulation as PHHs. This is also discussed in the discussion section (Line 589-591).